# Systemic Pulmonary Events Associated with Myelodysplastic Syndromes: A Retrospective Multicentre Study

**DOI:** 10.3390/jcm10061162

**Published:** 2021-03-10

**Authors:** Quentin Scanvion, Laurent Pascal, Thierno Sy, Lidwine Stervinou-Wémeau, Anne-Laure Lejeune, Valérie Deken, Éric Hachulla, Bruno Quesnel, Arsène Mékinian, David Launay, Louis Terriou

**Affiliations:** 1Department of Internal Medicine and Clinical Immunology, National Reference Centre for Rare Systemic Autoimmune Disease North and North-West of France, University of Lille, CHU Lille, F-59000 Lille, France; quentin.scanvion@chru-lille.fr (Q.S.); eric.hachulla@chru-lille.fr (É.H.); david.launay@chru-lille.fr (D.L.); 2Department of Haematology, Hôpital Saint-Vincent de Lille, Catholic University of Lille, F-59000 Lille, France; pascal.laurent@ghicl.net (L.P.); bruno.quesnel@chru-lille.fr (B.Q.); 3Internal Medicine Department, Armentières Hospital, F-59280 Armentières, France; t.sy@ch-armentieres.fr; 4Service de Pneumologie et ImmunoAllergologie, Centre de Référence Constitutif des Maladies Pulmonaires Rares, CHU Lille, F-59000 Lille, France; lidwine.wemeau@chru-lille.fr; 5Department of Thoracic Imagining, University of Lille, CHU Lille, F-59000 Lille, France; anne-laure.lejeune@univ-lille.fr; 6ULR 2694—METRICS: Évaluation des Technologies de Santé et des Pratiques Médicales, University of Lille, CHU Lille, F-59000 Lille, France; vdeken.chr@gmail.com; 7Department of Internal Medicine, AP-HP, Saint-Antoine Hospital, F-75012 Paris, France; arsene.mekinian@aphp.fr; 8INFINITE—Institute for Translational Research in Inflammation, University of Lille, F-59000 Lille, France; 9Inserm, U1286, F-59000 Lille, France

**Keywords:** interstitial lung disease, pneumonia, pleuritic effusion, pulmonary hypertension, pulmonary alveolar proteinosis, iatrogenic effects

## Abstract

Although pulmonary events are considered to be frequently associated with malignant haemopathies, they have been sparsely studied in the specific context of myelodysplastic syndromes (MDS). We aimed to describe their different types, their relative proportions and their relative effects on overall survival (OS). We conducted a multicentre retrospective cohort study. Patients with MDS (diagnosed according to the 2016 WHO classification) and pulmonary events were included. The inclusion period was 1 January 2007 to 31 December 2017 and patients were monitored until August 2019. Fifty-five hospitalized patients were included in the analysis. They had 113 separate pulmonary events. Thirteen patients (23.6%) had a systemic autoimmune disease associated with MDS. Median age at diagnosis of MDS was 77 years. Median time to onset of pulmonary events was 13 months. Pulmonary events comprised: 70 infectious diseases (62%); 27 interstitial lung diseases (23.9%), including 13 non-specific interstitial pneumonias and seven secondary organizing pneumonias or respiratory bronchiolitis–interstitial lung diseases; 10 pleural effusions (8.8%), including four cases of chronic organizing pleuritis with exudative effusion; and six pulmonary hypertensions (5.3%). The median OS of the cohort was 29 months after MDS diagnosis but OS was only 10 months after a pulmonary event. The OS was similar to that of the general myelodysplastic population. However, the occurrence of a pulmonary event appeared to be either an accelerating factor of death or an indicator for the worsening of the underlying MDS in our study. More than a third of pulmonary events were non-infectious and could be systemic manifestations of MDS.

## 1. Introduction

MDS are a heterogeneous clonal condition of pluripotent stem cells. They constitute the main haemopathy after 70 years of age. MDS are defined by chronic cytopaenia linked to qualitatively ineffective haematopoiesis [1,2,3,4]. It has been shown that 15 to 33% of myelodysplastic patients have associated autoimmune or auto-inflammatory diseases (ADs) [1,5,6]. Systemic vasculitis, connective tissue disease, inflammatory arthritis, neutrophilic pathologies [7] and, more rarely, autoimmune cytopaenia [8] have all been observed. Furthermore, there are systemic diseases that are incomplete and consequently more difficult to diagnose [7]. ADs isolated to a single organ have also been described [1,7,8]. MDS are now considered as complex disease and their clinical expression is not purely haematological.

Concerning pulmonary manifestations in MDS, data are scarce. Indeed, while the frequency of pulmonary events in the context of malignant haemopathies (all types combined) is commonly considered to be high and associated with a high morbidity and mortality [9] especially in acute leukaemias, chronic myeloid leukaemias, and lymphomas [10]. In the case of MDS, the data are less clear. Most lung diseases are attributed to infectious causes [10]. However, the other pneumonias, especially interstitial pneumonia, have rarely been studied [9,11]. The relative frequency of infectious versus non-infectious pneumonias in myelodysplastic patients has not been directly described. A better knowledge of the pulmonary events in these patients would allow a more effective and efficient diagnostic and therapeutic approach, by enabling the relevance of paraclinical examinations to be assessed.

The main purpose of our study was to describe all of the pulmonary events occurring in hospitalized patients with MDS, their type and their relative proportions. A secondary objective was to explore survival according to the type of event.

## 2. Materials and Methods

### 2.1. Patient Selection

We conducted a multicentre retrospective study. Patients with MDS (diagnosed according to the 2016 WHO classification) and pulmonary events were included at three centres in the north of France (Lille University Hospital, Lille Catholic University Hospitals, Armentières Hospital). The inclusion period was 1 January 2007 to 31 December 2017 and monitoring was carried out until August 2019. Patients were included if they had been hospitalized at least once for MDS and a pulmonary event (see online Supplementary Material S1). The exclusion criteria were cardiogenic acute oedema of the lungs, coding errors, lack of information and if MDS had worsened to acute myeloid leukaemia (AML) when the pulmonary event occurred.

### 2.2. Data Collection

Patients’ medical records were individually searched for (i) demographic data; (ii) comorbidities (smoking status: active, absent or weaned off tobacco for more than 3 years; the updated age-adjusted Charlson comorbidity index [12]; the Adult Comorbidity Evaluation 27 score, the Cumulative Illness Rating Scale for Geriatrics); (iii) MDS characteristics; (iv) molecular biology data by next-generation sequencing (NGS) (which improved during the inclusion period; the most recent NGS included an analysis of 36 genes); (v) overall survival (OS); (vi) characteristics of pulmonary events. During data collection, if a non-infectious pulmonary event was highlighted, a potentially iatrogenic treatment was sought in the patient’s history. The baseline considered was Pneumotox [13]. The pulmonary events considered were those diagnosed during or after hospitalization or diagnosed a posteriori during the analysis of medical records for this study. These diagnoses were based on the synthesis of clinical data, pulmonary function test, bronchoscopy and biological (including bronchoalveolar lavage analysis) and radiological studies. The CT scans were reviewed by an independent radiologist (specialized in thoracic imaging). We adopted a “dynamic integrated approach” as in the Update of the International Multidisciplinary Classification of the Idiopathic Interstitial Pneumonias [14]. The process of achieving a multidisciplinary diagnosis was dynamic, requiring confrontations between data from clinicians, biologists, radiologists and, when available, pathologists. In order to be retained, cases of interstitial lung diseases associated with MDS did not have to have an identified secondary aetiology.

### 2.3. Statistical Analysis

Results are expressed as median (interquartile range) [minimum–maximum] for continuous variables and as frequencies and percentage for categorical variables. The normality of the distributions was assessed graphically and using the Shapiro–Wilk test. The factors associated with mortality were studied by Cox models, including factors that varied during follow-up as time-dependent variables. Ordinal qualitative variables were subjected to trend tests. For specific analyses of pulmonary events, Cox models with adjustment to the International Prognostic Scoring System–Revised (IPSS–R) score were developed. The effects of each factor on survival were evaluated using the hazard ratio (HR) and the 95% confidence interval (95% CI). The statistical tests were all carried out with a bilateral risk α of 5%. Statistical analyses were performed using SAS software (version 9.4, SAS Institute Inc. Cary, NC, USA). All the statistical analyses were discussed and validated by a group of statisticians, represented by Mrs Valérie Deken.

### 2.4. Ethical Considerations

In accordance with French law, the creation of the database necessary for this study was declared beforehand to the Commission Nationale de l’Informatique et des Libertés (CNIL) and was included in the Lille University Hospital’s register of declared databases. The included patients had been informed beforehand in their letters of consultation or hospitalization that unless they expressed an objection, their medical data could be used in an anonymous form for medical research purposes. No objections were expressed by any of the included patients.

## 3. Results

### 3.1. Patients’ Characteristics

Fifty-five myelodysplastic patients with a total of 113 pulmonary events were included in the study. A flow chart of patient inclusion is presented in Figure 1. The characteristics of the patients are detailed in Table 1.

Regarding smoking status, seven patients (16.3%) were active smokers, 19 (44.2%) had been weaned off smoking for more than three years at the time of diagnosis of MDS and 17 patients (39.5%) had never smoked; the remaining 12 patients had missing data. Among the 55 patients, the “NGS MDS” molecular biological status was known for 22 patients (40%). At least one pathogenic mutation was confirmed in 19 of these 22 patients (86%). The lack of sufficient molecular data prevented us from carrying out statistical analyses of these data. Thirteen patients (23.6%) had an AD: two had polyarteritis nodosa (one with myocardial involvement), one had granulomatosis with polyangiitis, two had giant cell arteritis, one had hypocomplementemic urticarial vasculitis, one had unclassifiable large-vessel vasculitis, two had immune thrombocytopaenia (associated in one case with systemic lupus erythematosus), one had mixed connective tissue disease, one had psoriatic arthritis, one had unclassifiable auto-inflammatory disease and one had large granular lymphocyte leukaemia with systemic lupus erythematosus, anti-phospholipid syndrome and secondary Gougerot-Sjögren syndrome. Age at diagnosis of the various associated pulmonary events (all types combined) was 76 years (69–84) [29.0–92.0]. The time from diagnosis of MDS to each pulmonary event was 13 months (1–40) [−93.0–139.0]. Age at diagnosis of ADs was 69 years (42–71) [6.0–79.0]. This corresponded to a delay of −39 months (−139–0) [−361.0–16.0] between the diagnosis of MDS and ADs (which therefore mostly predated MDS diagnosis).

### 3.2. Pulmonary Events

There were 113 pulmonary events, including 70 infectious pneumonias (61.9%), 27 interstitial lung diseases (ILDs) (23.9%), 10 pleural effusions (8.8%) and six pulmonary hypertensions (5.3%) (see Figure 2). The presence of ADs did not significantly modify the proportions of pulmonary events (Fisher’s test, *p* = 0.88).

In 36 of the 70 infectious pneumonias (51.4%) the pathogen was undetermined (see Figure 3). For five of them (7.1%), the infectious causes initially retained by clinicians appeared questionable with retrospective knowledge of the evolution of the cases (treatments undertaken, subsequent events, etc.). A diagnosis of secondary organizing pneumonia (SOP) was suspected for these cases, without sufficient retrospective arguments to be accepted. When the pathogen was known, it was mainly bacterial (*n* = 26, 37.1%). We also found six fungal infections (8.6%), one case of mycobacterial infection (1.4%) and one case of identified viral infection (1.4%). The pathogens are detailed in Appendix A.

Regarding ILD (see Figure 4), data synthesis revealed the following patterns: non-specific interstitial pneumonias (NSIPs) (*n* = 13), some of which had a diffuse (*n* = 4) or focal (*n* = 3) fibrosing component; SOPs (*n* = 4); respiratory bronchiolitis–interstitial lung diseases in non-smoking patients (RB-ILD-ns) (*n* = 3); pulmonary alveolar proteinosis  (PAP) (*n* = 1); diffuse alveolar damage (*n* = 1). There was also a case in which retrospective analysis of the data was unable to distinguish between pulmonary vasculitis and SOP and a case in which the CT scan and clinical history were suggestive of PAP but without confirmation by staining with periodic acid-Schiff with bronchoalveolar lavage. Among the seven cases of fibrosis, four patients had no autoantibodies, two had positive anti-nuclear antibodies (indirect immunofluorescence on HEp2 cells) but without specific antibodies, and one had a weakly positive myositis dot-blot assay, but without significant specific antibody.

The 10 cases of pleural effusion comprised chronic organizing pleuritis with exudative effusion (COPE, *n* = 4), pleural calcification with exudate (*n* = 2), lymphocytic exudate (*n* = 1), lymphocytic transudate (*n* = 1), simple transudate (*n* = 1) and secondary pleurisy due to Escherichia coli (*n* = 1).

All six cases of pulmonary hypertension were precapillary and one also had a postcapillary component.

Figure 5 shows the monitoring of each patient with the different pulmonary events over time. Appendix A describe the treatments received for MDS. Our study was not designed to assess the impact of treatment on outcome and the heterogeneity of treatment precludes any firm conclusion.

### 3.3. Overall Survival

Thirty-five patients (64%) died during the study period (i.e., 12 years and 8 months). Median age at death was 79 years (72–85) [32.0–93.0]. The Kaplan–Meier survival function is presented in Figure 6. According to this, the median OS was 29 months (16–127) [1.0–184.0] after the diagnosis of MDS. The OS after pulmonary events (all types) was 10 months (1–22) [0.0–198.0]. AML occurred in six patients (11%) after 20 months (11–45) [11.0–52.0]. It was significantly predictive of death (*p* < 0.001). Five of the six patients died during the study period. Their post-AML survival was 5 months (2–7) [<1.0–8.0]. One patient was still alive 2 months after AML diagnosis (i.e., at the end of the study). The results of the univariate analyses of the predictive factors of post-MDS survival are available in Appendix A).

The secondary aim of the study was to study survival according to the different types of pulmonary events. For this purpose, we formed the following groups of analyses: Group A (“Infectious”): comprising the infectious pneumonias (excluding the five doubtful cases) and the bacterial pleurisy case (66 cumulative pulmonary events (58%)); Group B (“Organizing”): comprising the SOPs, the RB-ILD-ns cases and the COPE cases (12 cumulative pulmonary events (11%)); Group C (“NSIP”): comprising the NSIP cases (13 cumulative pulmonary events (11%)).

For the myelodysplastic patients in our cohort, according to a Cox model the diagnosis of a first infectious pneumonia (Group A) did not influence survival any differently from the diagnosis of Group B impairment or the diagnosis of NSIP (i.e., Group C) (see Table 2). The recurrence of at least one infectious pneumonia increased the hazard ratio versus the first infectious pneumonia.

### 3.4. Iatrogenic Data

The information collected shows that one of the NSIP cases with a focal fibrosing component occurred in a patient who had been treated for one year with methotrexate. A patient with chronic myelomonocytic leukaemia (CMML) had been treated with hydroxyurea; during his follow-up, one SOP and one COPE occurred. Finally, one case of NSIP, one case of SOP and the case of pulmonary vasculitis or SOP occurred during treatment with azacitidine.

## 4. Discussion

To our knowledge, this retrospective multicentre study is the first to describe the proportions of pulmonary events (both infectious and non-infectious) occurring in the myelodysplastic population and their relative effects on patient OS. The general characteristics of our cohort are close to those of the general myelodysplastic population [2,15]. Thus, myelodysplastic patients with pulmonary events do not seem to form a particular clinical population of MDS [11]. The distribution of different types of MDS was broadly consistent with that of previous international cohorts. However, we noted a tendency for an over-representation of multilineage dysplasia (as opposed to single lineage) and MDS with excess blasts. This finding is also described in MDS with associated systemic inflammatory and autoimmune manifestations [16], which represent 15% to 33% of MDS cases [1,5,6]. The representativeness of our cohort was strengthened by multicentre inclusion. Indeed, the three inclusion centres have a varied patient base. Lille University Hospital is a national reference centre for systemic diseases. Armentières Hospital is a rural hospital. Lille Catholic University Hospitals are city hospitals. However, the retrospective nature of the data did not allow us to overcome all selection and information biases.

MDS are chronic pathologies that usually induce an immunocompromised status. Up to 60% of patients with neutropaenia (whatever the cause) may develop a pulmonary infiltrate during follow-up [17]. In practice, most MDS lung diseases are therefore attributed to infectious causes [9]. However, in immunocompromised patients or patients with cancer (not specifically MDS), some authors estimate that a quarter to a half of infiltrates would be non-infectious [18]. Our results support this since infectious pneumonias represented at most only 61.9% (i.e., 70/113) of pulmonary events. Furthermore, this proportion may have been overestimated because it is common for pneumonia to be diagnosed, even in the absence of an identified pathogen. However, we noticed that there were some patients whose pneumonias were not improved by several lines of antibiotic therapy, except when it was associated with corticosteroids. These cases could therefore be unrecognized SOP. However, we only included hospitalized patients; consequently, less severe pneumonias occurring at home were not counted. Our results are therefore not applicable to outpatient MDS patients

Twenty-four percent of pulmonary events had patterns of ILD. These pulmonary manifestations have been little evaluated apart from a few case reports or case series. A retrospective study of 827 myelodysplastic patients at the Mayo Clinic found 18 (2%) ILDs, including 44% NSIP (half of which had a fibrosing component) and 22% organizing pneumonias [11]. Our study confirmed this relative distribution of patterns, with 48% NSIP and 30% SOP and RB-ILD-ns. However, we did not confirm the previously reported high prevalence of 5q- abnormalities in MDS patients [11], with only two cases in our study. It should be noted that the presence of ILD predisposes to infections, especially fungal infections. In our cohort, an NSIP pre-existed for four out of six fungal infections. On the other hand, for 32 clinical and radiological suspicions of invasive pulmonary aspergillosis, Kim et al. showed that a lung biopsy corrected the diagnosis in favour of SOP in seven cases (22%) [19]. This supports our suspicion of underdiagnosis of SOP. Between the hypotheses of a fortuitous association or systemic pulmonary manifestations associated with MDS, the literature provides several arguments in favour of the latter. Organizing pneumonia (OP) is an inflammatory pathology of the lower respiratory tract. A distinction is made between cryptogenic and secondary forms. Secondary aetiologies are multiple. Among them we found malignant haemopathies such as acute lymphomas and leukaemias, but MDS are not often cited [20,21,22,23]. However, in line with our results, case series of OP induced by MDS have been described [24,25,26,27,28,29,30,31,32,33,34]. The first case was published in 1990 [35]. From a pathophysiological point of view, in this pulmonary involvement, inflammatory cells infiltrate the alveolar interstitial tissue after any primary endothelial lesion [21]. Since the neutrophils and eosinophils are dysplastic in MDS, these appear to be a probable aetiology of OP. Furthermore, Sweet’s syndrome (SS) is another strong argument for a causal association between MDS and OP. Indeed, the malignant form (as opposed to the idiopathic form) of SS is a paraneoplastic syndrome and is particularly induced by myeloid haemopathies (dermatosis can happen before, during or following the haemopathy) [33,36,37,38]. It is a neutrophilic dermatosis. However, SS frequently associates with cutaneous involvement, a secondary OP [37,38,39,40]. Its pulmonary involvement is one of the most frequent and most studied systemic neutrophilic infiltrates. It occurs during dermatosis, but can be inaugural and therefore occurs alone [41]. On the other hand, a particular histological form, called histiocytoid Sweet syndrome by some authors [42] myelodysplasia cutis by others [43] is described. In this form, the tumour cells have a combined myeloid and monocytic phenotype and CD3+ T cells are also present. This form of SS is said to be a cutaneous MDS. Given the existence of only cutaneous SS associated with MDS and myelodysplasia cutis, it is possible to consider only pulmonary SS associated with MDS or “myelodysplastic organizing pneumonia”. In addition, the signature profiles of cytokines in myelodysplastic patients confirm higher levels of TNF-α, IFN-γ, TGF-β, IL-6 and IL-18 [44]. Similarly, subjects with interstitial lung disease were found to have higher levels of IL-6 [45], TNF-α [46], TGF-β [47], IFN-γ [48] and other inflammatory cytokines. Thus the link of interstitial lung disease with MDS could be due to immune deregulation [11]. In addition, myelodysplastic patients have higher G-CSF levels [44] and drug-induced SS with OP occurs most often after treatment with G-CSF [36]. We note another element of link. Indeed, in our patients (as in other case reports) a close linked evolution between interstitial lung diseases and MDS was observed. In view of all the above arguments, we propose to classify and consider OP in myelodysplastic patients as secondary OP (and not cryptogenic). It is also sometimes proposed that interstitial lung diseases are caused by pulmonary leukocytoclastic vasculitis [49,50,51,52], associated with systemic autoimmune or auto-inflammatory diseases [21] The pulmonary involvement then constitutes only a part of systemic vasculitis, itself associated or not with MDS. In our cohort, this pulmonary vasculitis attack was strongly suspected for a pulmonary manifestation occurring in patient No. 41, who also had a severe hypocomplementemic urticarial vasculitis (with amputation of the lower limbs).

Pulmonary alveolar proteinosis (PAP) is a rare condition characterized by the alveolar accumulation of components of the surfactant. Autoimmune PAP (anti-G-CSF serum antibodies) represents 90% of PAP cases [53,54]; the remaining cases are secondary (8%) or hereditary (2%). Among secondary PAPs, more than 3/4 are caused by MDS, and therefore 6% of PAPs in total [53,54]. They are due to a qualitative or quantitative macrophages dysfunction for the clearance of the surfactant [53,54,55]. GATA2 haploinsufficiency has been described as a possible aetiology of secondary MDS and PAP [56,57]. When MDS ± PAP are associated with mycobacterial infections, disseminated HPV (human papilloma virus) or opportunistic fungal infections, and with lack of monocytes, B cells and NK cells, they constitute the MonoMAC syndrome [58]. This is the case in our patient No. 55: multilineage MDS + PAP + pneumonia caused by Mycobacterium kansasii + NSIP. GATA2 haploinsufficiency is also associated with pulmonary hypertension [59].

Concerning the iatrogenic effects, apart from the possible involvement of methotrexate in a case of fibrosing NSIP, there is no strong argument in favour of other iatrogenic treatment effects. Indeed, the toxic hypothesis of hydroxyurea must be put into perspective. Firstly, in our cohort, pleurisy and SOP occurred, respectively 23 and 29 months after the end of treatment. Secondly, in the literature, only two cases (one per complication [60,61]) have been reported and they clearly lack arguments of accountability or suffer from confusion bias. For example, hydroxyurea was concomitant with IFN-γ and cytarabine, which are themselves listed as possible causes of SOP [13]. Furthermore, azacitidine was suspected in about ten case reports describing different pulmonary events [13]. However, this treatment is indicated for CMML and AML, and for MDS with an intermediate to poor prognosis. Independently of azacitidine, all these pathologies can themselves be associated with ILD. Prior MDS treatment and ADs seemed to play no significant role in ILD development.

Nine non-purulent pleural effusions occurred, accounting for 8% of pulmonary events in the cohort. This relatively large proportion opens a new perspective on the pleural complications of MDS, which have been very little studied. Indeed, although pleural effusions are frequently associated with lymphomas, they are only rarely discussed for MDS [62,63]. Four cases of COPE were observed, in 4 out of 55 patients. COPE is a benign mesothelial proliferation, with varying degrees of more or less fibrous reactive hyperplasia, when the pleura is injured [64]. This is the first time that the MDS and COPE association has been described. Two patients had at least one severe respiratory infection in their history. However, for the other two patients, no aetiology was highlighted.

Pulmonary hypertension is an initially insidious and paucisymptomatic pathology, and this is especially problematic when associated with MDS, as dyspnoea can be attributed to anaemia. Unlike myeloproliferative syndromes, there are few data for MDS (the pathophysiological link has not yet been clearly established). There are three case reports [65,66], and a recent prospective study found three cases of pulmonary hypertension in 36 myelodysplastic patients (8%) [67].

The median OS was similar to that of the general myelodysplastic population [2]. MDS with pulmonary events does not appear to be a distinct subgroup of MDS in terms of survival. However, pulmonary events are significant events in the history of the disease. In our study, pulmonary events occurred towards the end of the disease history (median 10 months before death). Thus, the occurrence of a pulmonary event appears to be a marker of severity during the natural history of MDS. However, the retrospective nature of the study must be kept in mind as a source of potential bias if the start or end of follow-up was not carried out in our inclusion centres. This study did not allow us to determine a relative risk of death for MDS patients with pulmonary events versus those without. However, it is interesting to note that one type of pulmonary event did not decrease survival more significantly than another (i.e., similar HR CI 95% in the three groups, A, B and C). Nevertheless, it should be borne in mind that the effect of the complications in Groups B and C may have been underestimated by our model because of a smaller number of cases (a lack of power cannot be excluded). We also noted a logical cumulative effect, since the recurrence of at least one case of infectious pneumonia in a given patient increased the HR. In addition, it should be remembered that our study focused on information from hospital medical records. Thus, the known infectious complications were more severe than ambulatory pneumonia. The presence of ADs had no significant effect on survival. The effect of this parameter on the OS of MDS patients is discordant with various other studies [1]. Nevertheless, in a series of 123 myelodysplastic patients with an associated autoimmune or auto-inflammatory pathology versus 665 patients without, the survival did not differ [16].

## 5. Conclusions

In conclusion, a wide variety of pulmonary complications can occur in patients with MDS. More than a third of them could correspond to systemic pulmonary manifestations, comprising several ILDs, pleural effusions and pulmonary hypertension. Future prospective studies will be necessary to assess the risk of developing a pulmonary complication and to explore appropriate therapeutic strategies (corticosteroid therapy, hypomethylating treatments, etc.).

## Figures and Tables

**Figure 1 jcm-10-01162-f001:**
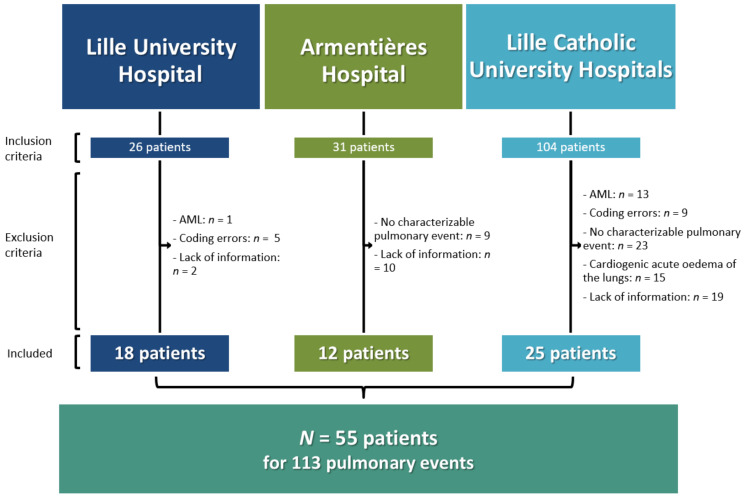
Patient inclusion flow chart. (AML: MDS had worsened to acute myeloid leukaemia when the pulmonary event occurred).

**Figure 2 jcm-10-01162-f002:**
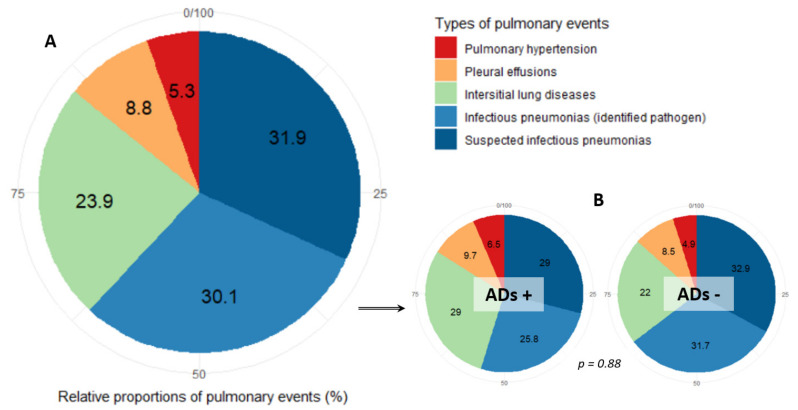
Pie charts of the different types of pulmonary events occurring in myelodysplastic patients. (**A**): Overall cohort. (**B**): myelodysplastic syndrome with associated autoimmune or auto-inflammatory diseases (ADs +) versus without (ADs −).

**Figure 3 jcm-10-01162-f003:**
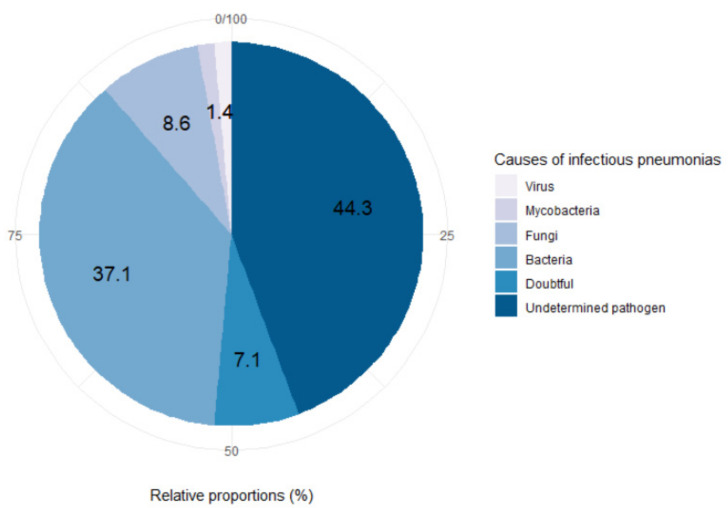
Pie chart of the causes of infectious pneumonias occurring in myelodysplastic patients.

**Figure 4 jcm-10-01162-f004:**
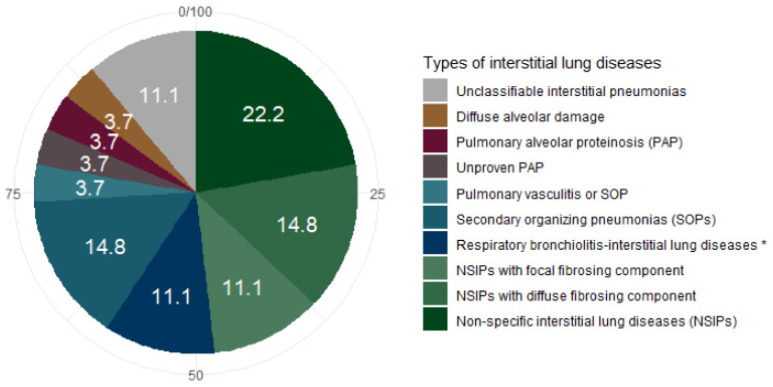
Pie chart of the patterns of ILDs in myelodysplastic patients. (* in non-smoking patients).

**Figure 5 jcm-10-01162-f005:**
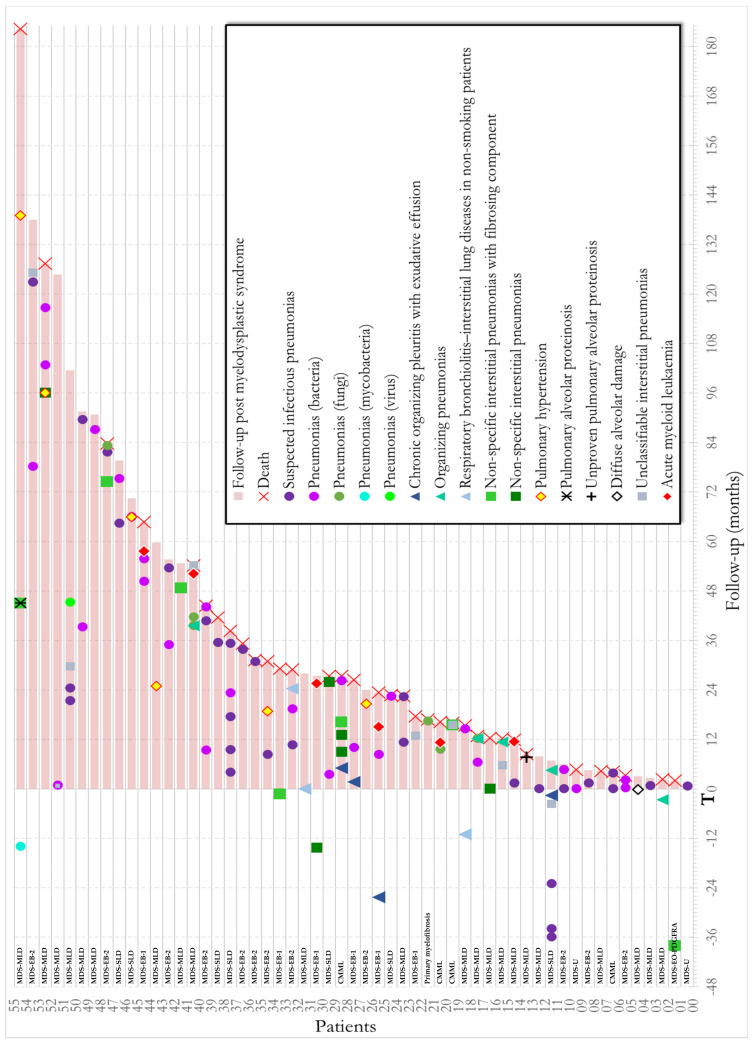
Historiogram of the cohort: chronology of pulmonary events in myelodysplastic patients. (T_0_: diagnosis of myelodysplastic syndrome).

**Figure 6 jcm-10-01162-f006:**
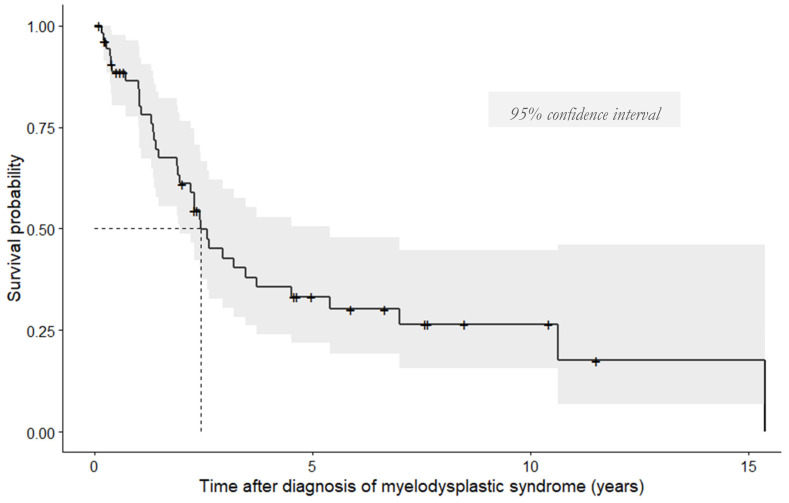
Kaplan–Meier survival curve for our cohort.

**Table 1 jcm-10-01162-t001:** General characteristics of patients.

	Lille University Hospital (*n* = 18)	Armentières Hospital (*n* = 12)	Lille Catholic University Hospitals (*n* = 25)	Total (*n* = 55)	
Gender ratio (M/F)	13/5 = 2.6	7/5 = 1.4	23/2 = 11.5	43/12 = 3.6	
Median age at diagnosis of MDS	69 (50–73) [29.0–87.0]	80 (78–82) [64.0–88.0]	78 (69–83) [56.0–92.0]	77 (66–83) [29.0–92.0]	
MDS or MDS/MPN (WHO 2016)					
MDS-SLD	1	2	4 ^§^	7	(52.7%)
MDS-MLD	10	2 ^¤^	10	22	
MDS-EB-1	0	3	3 ^¤^	6	(32.7%)
MDS-EB-2	3	1	8 ^¤^	12	
MDS-U	2	0	0	2	(3.6%)
CMML	1	3	0	4	(7.3%)
MDS-EO-PDGFRA	1	0	0	1	(1.8%)
Primary myelofibrosis	0	1	0	1	(1.8%)
MDS					
Primary	15	11	20	46	(83.6%)
Secondary Cancer chemotherapy					
	1	1	4	6	(16.3%)
Exposure ^¶^	2	0	1	3	
IPSS				*N* = 50	
Good	4	1	4	9	(18%)
Intermediate 1	6	3	6	15	(30%)
Intermediate 2	5	1	11	17	(34%)
Poor	1	0	2	3	(6%)
*NA*	*1*	*3*	*2*	*6*	*(12%)*
IPSS-R				*N* = 50	
Very good	3	1	2	6	(12%)
Good	4	0	5	9	(18%)
Intermediate	3	3	7	13	(26%)
Poor	2	0	3	5	(10%)
Very poor	4	1	6	11	(22%)
*NA*	*1*	*3*	*2*	*6*	*(12%)*
CPSS				*n* = 4	
Good	0	0	0	0	
Intermediate 1	1	0	0	1	(25%)
Intermediate 2	0	1	0	1	(25%)
Poor	0	1	0	1	(25%)
*NA*	0	1	0	*1*	*(25%)*
Comorbidities					
Charlson index^1^	5 (3–6) [0.0–8.0] ^&^	6 (5–7) [2.0–9.0]	7 (4–7) [1.0–16.0]	6 (4–7) [0.0–16.0] ^&^	
CIRS-G	9 (5–11) [0.0–18.0] ^&^	8 (6–9.5) [3.0–12.0]	9 (7–10) [2.0–19.0]	9 (7–10) [0.0–19.0] ^&^	
ACE-27	2 (2–3) [0.0–3.0] ^&^	2.5 (2–3) [1.0–3.0]	3 (2–3) [0.0–3.0]	3 (2–3) [0.0–3.0] ^&^	
ADs	12	0	1	13	(23.6%)

WHO: World Health Organization classification of myeloid neoplasms 2016; IPSS(-R): (Revised) International Prognostic Scoring System; NA: missing data; CPSS: CMML-specific Prognostic Scoring System; CIRS-G: Cumulative Illness Rating Scale for Geriatrics; ACE-27: Adult Comorbidity Evaluation-27; ADs: associated autoimmune or autoinflammatory diseases. ^§^ and ^¤^: including, respectively 3 and 1 cases with ring sideroblasts. ^¶^: occupational exposure to benzene, hydrocarbons or heavy metals. ^&^: 1 missing data. ^1^: the updated age-adjusted Charlson index, according to Bannay et al. 2016 Med. Care [12].

**Table 2 jcm-10-01162-t002:** Comparison of predictive effects of the three main groups of pulmonary events (Infectious, SOP, NSIP) on survival, according to a multivariate time-dependent Cox model.

Pulmonary Events		Patient Months	Deaths	HR	[HR 95% CI]
Group A (“Infectious”):					
1 event		14,644	15	5.6	[2.2–14.3]
≥2 events		5471	11	8.6	[2.8–26.3]
No event		43,421	9		
Group B (“Organizing”):	Yes	2991	6	4.5	[1.7–12.0]
	No	60,545	29		
Group C (“NSIP”):	Yes	6655	7	5.8	[1.6–20.9]
	No	56,881	28		

HR: hazard ratio adjusted for IPSS-R score; HR 95% CI: 95% confidence interval of the hazard ratio.

## Data Availability

The data presented in this study are available on request from the corresponding author. The data are not publicly available due to the included patients have consented to the use of their medical data in anonymous form for medical research purposes, but not to its access in a publicly accessible repository.

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
