# Peer review of "Systemic Pulmonary Events Associated with Myelodysplastic Syndromes: A Retrospective Multicentre Study"

_jcm, 2021, doi:10.3390/jcm10061162_

Round 1
Reviewer 1 Report
The authors describe pulmonary manifestations retrospectively in a cohort of patients with MDS. Although I agree with the authors that inflammation plays a significant and yet under-investigated role in MDS, I have major comments:
- this retrospective analysis includes only a small number of patients with MDS and evaluates pulmonary manifestations only in a hospitalized cohort. This could not be changed but limits the findings significantely to a predefined subgroup of MDS and could not make conclusions to all MDS patients. In the study, the bias is really a main issue. I wonder why the authors did not include outpatient MDS patients and/or patients without pulmonary problems as a control?.
- The diagnosis of pulmonary findings was not substantiated by bronchoscopy or histopathology in the majority of patients which very much weakens the data and the realiability of the findings. The authors themselve use terms such as suspicious, suggestive (this is not evidence)
- No account at all was made to treatments received for MDS!! What was their impact on outcome and on the pulmonary disorders??
- I do not see a point in including NGS data on the small number of patients without mentioning a possible correlation of these mutations to lung disorders
- The statistical validity of dividing a small group into three further subgroups is questionable.
- The data do not allow the conclusions made in the first part of the discussion.
- What about cytokines? were they measured in any patient? If not, then the lengthy discussion on these cytokines is redundant. The same is true for Sweat Syndrome
Minor comments:
- CPSS for only 4 patients?
- In table 1, there are two ranges after the median? The content of the brackets needs to be explained
- Figures 2,3,4 and 5 are repetitive and could be less and more concise
Author Response
Point-by-point response to Reviewer #1
The authors describe pulmonary manifestations retrospectively in a cohort of patients with MDS. Although I agree with the authors that inflammation plays a significant and yet under-investigated role in MDS, I have major comments:
- this retrospective analysis includes only a small number of patients with MDS and evaluates pulmonary manifestations only in a hospitalized cohort. This could not be changed but limits the findings significantely to a predefined subgroup of MDS and could not make conclusions to all MDS patients. In the study, the bias is really a main issue. I wonder why the authors did not include outpatient MDS patients and/or patients without pulmonary problems as a control?.
>> We thank the reviewer for this analysis.
Indeed, the inflammatory system damage caused by MDS is currently under-investigated. Our study is part of an innovative approach to integrate pulmonary manifestations as related features of MDS.
No study to our knowledge has observed MDS through the prism of all pulmonary manifestations. Few studies have focused on any one manifestation, and none has focused on all of them.
Before we could undertake a more ambitious prospective study, we needed to retrospectively look for strong evidence in favour of the diversity of pulmonary events associated with MDS. This study was therefore conducted retrospectively. We limited our study to hospitalized patients for two main reasons: inclusion screening and the density of available medical information. Although the analysis of outpatient MDS patients would certainly have been interesting, they are unfortunately difficult to include. Indeed, their available medical information is too insufficient to allow a reliable diagnosis of the pulmonary manifestations (biological results and scans not accessible or limited because they are external, evolution not known, etc.). We therefore chose to restrict the study to the hospitalized population, to avoid great analytical bias. Moreover, we acknowledge the limit reported by Reviewer #1 (see last line p.9). We have added a sentence to support this known limit further.
Nevertheless, although we have only included hospitalized patients, the general characteristics of our population are very close to those of the largest published MDS cohorts. Our cohort therefore appears to be close to the general myelodysplastic population. We develop these elements in the first paragraph of the discussion p.9). The previously published data from large MDS cohorts constitute the control population without pulmonary manifestation.
Concerning the total number of patients included (55), this should not be considered as small, if we keep in mind that we are studying unsystematic complications in the course of an already relatively infrequent pathology (incidence <5/100,000 in the general population) [1]. Moreover, our cohort is larger than that of a retrospective study by the Mayo Clinic (interstitial lung diseases/MDS) [2].
- The diagnosis of pulmonary findings was not substantiated by bronchoscopy or histopathology in the majority of patients which very much weakens the data and the realiability of the findings. The authors themselve use terms such as suspicious, suggestive (this is not evidence).
>> We must make an important clarification. The results of bronchoscopies and bronchioloalveolar washings were integrated for the diagnosis of pulmonary findings, when they were available.
Concerning histopathology, it is true that it was absent in several patients. For interstitial lung diseases, histopathology is the historical gold standard. Nevertheless, due to the complexity and the risks involved in lung biopsies, they are no longer systematically performed.
In order to be retained, cases of interstitial lung diseases associated with MDS did not have to have an identified secondary aetiology. For our study, we adopted a "dynamic integrated approach" as in the Update of the International Multidisciplinary Classification of the Idiopathic Interstitial Pneumonias [3]. The process of achieving a multidisciplinary diagnosis was dynamic, requiring confrontations between data from clinicians, biologists, radiologists and, when available, pathologists. The biopsy is not always more informative than high-resolution computed tomography (HRCT).
We have changed the Materials and Methods section to clarify these points and we have modified the ambiguous terms in the Results and Discussion sections.
We thank the reviewer #1 for this comment.
- No account at all was made to treatments received for MDS!! What was their impact on outcome and on the pulmonary disorders??
>> During the retrospective data collection, the treatments received for MDS were recorded. For example, 16 patients received azacitidine, 1 received lenalidomide, 1 received ASCT , etc.
Due to constraints on the maximum number of words, we were unable to include details of these treatments in the manuscript. Moreover, the objective of our study was epidemiological and not about treatments.
We propose a supplemental table describing these treatments. Our study was not designed to assess the impact of treatment on outcome and the heterogeneity of treatment precludes any firm conclusion.
- I do not see a point in including NGS data on the small number of patients without mentioning a possible correlation of these mutations to lung disorders
>> During data collection, we were careful to collect NGS results to explore a possible link between a given mutation and a type of pulmonary event. Unfortunately, the lack of sufficient molecular data did not allow us to carry out statistical analyses of these data (it should be remembered that the study is retrospective).
We have added a sentence to take this comment into account.
- The statistical validity of dividing a small group into three further subgroups is questionable.
>> All the statistical analyses were discussed and validated by a group of statisticians, represented by Mrs Valérie Deken in the list of authors of the manuscript.
- The data do not allow the conclusions made in the first part of the discussion.
>> This section compares our cohort results (general characteristics of our cohort) with the general characteristics of large MDS cohorts already published in the literature and widely documented (cited in references 1, 2, 5, 6, 11, 15, 16). We simply note that the characteristics of our cohort do not present any major divergence.
We remain cautious about interpretations ("do not seem to form", "broadly", "tendency"). We also point out the biases of a retrospective study in the discussion section.
- What about cytokines? were they measured in any patient? If not, then the lengthy discussion on these cytokines is redundant. The same is true for Sweat Syndrome
>> Cytokines were not measured in the cases of our retrospective cohort (non-interventional study).
The part of the discussion described by Reviewer #1, merely details finding in the literature that provide several arguments in favour of a physiopathological link between MDS and ILD.
Our study describes the set of pulmonary events in MDS. We believe it is important to detail the arguments in the literature distinguishing the hypotheses of fortuitous or physiopathological association.
Minor comments:
- CPSS for only 4 patients?
>> CPSS is the CMML-specific Prognostic Scoring System. Therefore, only the 4 CMML cases are concerned.
- In table 1, there are two ranges after the median? The content of the brackets needs to be explained
>> Results are expressed as median (interquartile range) [minimum–maximum] for continuous variables (see first line of paragraph 2.3.).
- Figures 2,3,4 and 5 are repetitive and could be less and more concise
>> The main purpose of our study was to describe all of the pulmonary events occurring in patients with MDS, their type and their relative proportions. This is why we have detailed these figures. Each of them provides different information (details of sub-types of events, chronology of events, etc.). We would prefer to keep these figures, in accordance with the comments of reviewers 2 and 3.
References
- Ma, X. Epidemiology of Myelodysplastic Syndromes. Am J Med 2012, 125, S2–S5, doi:10.1016/j.amjmed.2012.04.014.
- Nanah, R.; Zblewski, D.; Patnaik, M.S.; Begna, K.; Ketterling, R.; Iyer, V.N.; Hogan, W.J.; Litzow, M.R.; Al-Kali, A. Deletion 5q Is Frequent in Myelodysplastic Syndrome (MDS) Patients Diagnosed with Interstitial Lung Diseases (ILD): Mayo Clinic Experience. Leukemia Research 2016, 50, 112–115, doi:10.1016/j.leukres.2016.10.002.
- Travis, W.D.; Costabel, U.; Hansell, D.M.; King, T.E.; Lynch, D.A.; Nicholson, A.G.; Ryerson, C.J.; Ryu, J.H.; Selman, M.; Wells, A.U.; et al. An Official American Thoracic Society/European Respiratory Society Statement: Update of the International Multidisciplinary Classification of the Idiopathic Interstitial Pneumonias. Am J Respir Crit Care Med 2013, 188, 733–748, doi:10.1164/rccm.201308-1483ST.
Reviewer 2 Report
This is a well-conducted study with compelling preliminary data on pulmonary complications occurring during Myelodysplastic syndrome. Overall this study deserves fully considerations for pubblication.
I have just few comments that could help it to make more focused on :
- In figure 5 the type of myelodysplastic syndrome should be described;
- No mention on the specific role played by each treatment on pulmonary complications development is made. I do think that a mention of effects of each drugs on observed complication worth to be included;
- Authors conclude that pulmonary events do not affect OS of these patients, but surprisingly they assert that such event predicts poor hematological response. Such statements look to be quite questionable, please clarify;
- A mention of NGS is done but no comment on specific genes analyzed is included, nor any correlation with pulmonary events onset. If this analysis has not been performed I would suggest to remove it.
Reviewer 3 Report
In this paper, the Authors describe the incidence of pulmonary events in MDS patients, showing the high frequency of how non-infectious events. The onset of a pulmonary event appears to be an accelerating event.
The paper is well written and the results are clear.
In my view, the main limitation is the relatively small size of the cohort, however, the patients included are very well described.
Minor: Please check Figure 6: In the text it is stated that median OS is 29 months, this is not consistent with the figure, where the median survival is less than 5 months. I think that probably the legend for x axis should be corrected in "Time after diagnosis of myelodisplatisc syndrome (years)"
Round 2
Reviewer 1 Report
The authors did their best to address the concerns raised as far as they could.
Two wordings in the abstract:
a) for clarification, the word hospitalized needs to be added in the sentence: "Fifty-five patients were included in the analysis".
b) The final sentence "More than a third of pulmonary events were systemic manifestations of MDS and non-infectious." needs to be rephrased, ex: More than a third of pulmonary events were non-infectious and could be systemic manifestations of MDS.
